# Ulceroprotective Effects of *Epilobium angustifolium* Extract in DSS-Induced Colitis in Mice

**DOI:** 10.3390/cimb47060444

**Published:** 2025-06-10

**Authors:** Rumyana Simeonova, Rositsa Mihaylova, Reneta Gevrenova, Yonko Savov, Dimitrina Zheleva-Dimitrova

**Affiliations:** 1Department of Pharmacology, Pharmacotherapy and Toxicology, Faculty of Pharmacy, Medical University of Sofia, 1000 Sofia, Bulgaria; rmihaylova@pharmfac.mu-sofia.bg; 2Department of Pharmacognosy, Faculty of Pharmacy, Medical University of Sofia, 1000 Sofia, Bulgaria; rgevrenova@pharmfac.mu-sofia.bg; 3Institute of Emergency Medicine “N. I. Pirogov”, Bul. Totleben 21, 1000 Sofia, Bulgaria; yonko_savov@hotmail.com

**Keywords:** *Epilobium angustifolium*, DSS-induced ulcerative colitis, oxidative stress, colon inflammation

## Abstract

Ulcerative colitis (UC) is a chronic inflammatory bowel disease (IBD) associated with recurrent inflammation and ulceration of the colonic mucosa. Conventional treatments, including corticosteroids, have significant side effects, driving the need for safer, effective alternatives. The present study aimed to investigate the mitigating effects of *Epilobium angustifolium* extract (EAE) in a Dextran sulfate sodium (DSS)-induced colitis mouse model, in a comparative manner to the reference drug dexamethasone (DXM). The severity and progression of colitis were evaluated through disease activity indices and a range of inflammatory and oxidative stress markers, assessed using multiple analytical methods. EAE treatment significantly reduced colonic inflammation, as indicated by decreased myeloperoxidase (MPO) activity, lower levels of malondialdehyde (MDA), and reduced white blood cell counts. EAE also enhanced antioxidant defenses, increasing glutathione (GSH) levels by 64%, and boosting catalase (CAT), superoxide dismutase (SOD), and glutathione peroxidase (GPx) activities by 36%, 53%, and 70%, respectively. Histopathological analysis confirmed EAE’s efficacy in attenuating colonic injury and inflammation. The blood parameters hemoglobin, erythrocytes, and hematocrit were also improved. Our study shows that EAE has potential as a natural therapeutic candidate for the treatment of UC, demonstrating efficacy comparable to that of conventional pharmacological treatments.

## 1. Introduction

Inflammatory bowel diseases (IBDs), including Crohn’s disease and ulcerative colitis, are chronic, remitting inflammatory disorders of the gastrointestinal tract with complex etiologies involving genetic predisposition, environmental triggers, dysbiosis, and aberrant immune responses [1,2]. The resulting symptoms, which include a variety of severe symptoms, such as diarrhea, rectal bleeding, and abdominal pain, inevitably affect the quality of life of patients. The disease is usually relapsing, including periods of remission and exacerbation, with abdominal cramps, bloody diarrhea, and frequent manifestations of inflammation limited to the mucosal layer of the colon [3].

A substantial proportion of patients with UC were found to experience disease progression within a follow-up period of less than 10 years. Despite the availability and earlier use of advanced therapeutic options, disease progression was observed to occur sooner in patients diagnosed within the last decade compared to those diagnosed before 2014 [4].

The incidence and prevalence of this disease continue to increase, leading to ever-increasing healthcare and therapy costs. Treatment of UC is primarily based on disease severity and aims to induce and maintain long-term remission. Current treatment options range from conventional first-line therapy (corticosteroids, 5-aminosalicylates, and immunomodulators such as azathioprine) to targeted biological therapies with monoclonal antibodies and oral small molecules, such as tyrosine kinase inhibitors [5,6]. Unfortunately, both types of therapeutic strategies are associated with numerous adverse effects and compromise of treatment due to non-adherence to the respective therapy [3]. Up to 30% of UC patients have primary non-response to biologics or small molecules [7,8]. Additionally 20–37% of patients lose response to biologics over time, with immunogenicity being the leading contributor [9].

A hallmark of IBD pathogenesis is the loss of intestinal immune homeostasis, characterized by excessive activation of both innate and adaptive immune pathways, increased infiltration of leukocytes, and sustained production of proinflammatory cytokines and chemokines [1,10]. These processes lead to disruption of epithelial barrier integrity, tissue remodeling, and a heightened risk of extraintestinal complications, including metabolic and thrombotic disorders [11,12]. Despite advances in understanding disease mechanisms, current therapeutic options—including corticosteroids, immunosuppressants, and biologics—are often associated with limited long-term efficacy, adverse side effects, and risk of systemic immunosuppression [5].

Among the widely used experimental models for investigating IBD pathophysiology is the Dextran sulfate sodium (DSS)-induced colitis model in rodents. DSS exposure induces epithelial injury, immune cell activation, and cytokine-driven inflammation, closely resembling the clinical and histopathological features of human ulcerative colitis [13]. This model is particularly valuable for evaluating therapeutic candidates and their ability to modulate immune responses at both the mucosal and systemic levels [14].

The discovery of therapeutic agents that are both highly effective and have minimal side effects remains a challenging aspect of treating UC. The development of novel drugs and improvement of the treatment strategy by implementing personalized medicine are warranted to achieve optimal disease control [15].

Natural products and phytochemicals have gained significant interest as complementary or alternative treatments for chronic inflammatory diseases, leading to numerous scientific investigations [16]. One such candidate is *Epilobium angustifolium* (commonly known as fireweed), a medicinal plant traditionally used in various folk remedies for gastrointestinal, urinary, and inflammatory ailments [17]. *E. angustifolium* is a perennial herb with creeping rhizome. The stem is 50 to 150 cm high, simple, rarely branched, and often reddish. The leaves are thin, usually alternate, oblong-lanceolate, 4–15 cm long, and 0.4–3.5 cm wide, with convex, reticulately branched pale lateral veins below, sessile, or on short petioles. The inflorescence is a long raceme. The flowers are slightly irregular; petals 10–15 mm long and 5–8 mm wide and violet-pink. The capsule is 4–8 cm long, often reddish. The seeds are pale brown, oblong, and narrowed at both ends [18]. Preliminary studies have attributed its biological activity to the abundant presence of polyphenolic constituents, including flavonoids, phenolic acids, and ellagitannins known to interfere with oxidative stress, NF-κB signaling, and inflammatory cytokine production [17,19,20]. Previous investigations on the *E. angustifolium* aerial parts allowed for dereplication/annotation of 121 specialized metabolites by means of ultra-high-performance liquid chromatography—high resolution mass spectrometry [21]. The methanol-aqueous extract exerted antioxidant activity in radical scavenging, reducing power and metal chelating assays. The extract also had a distinct impact on α-glucosidase (3.48 mmol ACAE/g) and moderate activity towards α-amylase (0.44 mmol ACAE/g) and lipase (8.03 OE/g). It inhibited acetyl- and butyrylcholinesterase (2.05 and 1.67 mg GALE/g) and had a prominent anti-tyrosinase effect (61.91 mg KA/g). The polyphenolic profiling described in the aforementioned study was delineated by flavonoids (deoxyhexosides and hexuronides of kaempferol and quercetin), gallo- and ellagitannins (oenothein B, ellagic acid, galloyl-HHDP-hexose, and ellagic acid O-pentoside), and acylquinnic acids (chlorogenic, neochlorogenic, and 3-*p*-coumaroylquinic acid) [21]. Recently, the gastroprotective effect of oenothein B, with a particular emphasis on its ability to modulate various factors, including oxidative stress, prostaglandins, and gastric acid secretion was revealed. Animals pretreated with oenothein B showed a significant reduction in volume (mL) and total acidity, indicating that part of oenothein B’s gastroprotective effect may involve systemic actions, even when administered directly into the duodenum [22]. In addition, the antiulcer activities of elagic acid were evaluated in acute (ethanol and indomethacin) and chronic (acetic acid) ulcer models in Wistar rats. It was found that ellagic acid exerts antiulcer activity by strengthening the defensive factors and attenuating the offensive factors [23]. However, comprehensive in vivo data elucidating the therapeutic relevance of *Epilobium* extracts, a rich source of ellagitannins in the setting of colitis, remain limited. The purpose of the present study was to evaluate the ulceroprotective effects of EAE in mice with DSS-induced colitis and to compare these effects with the positive control dexamethasone, a widely used corticosteroid in the clinical practice.

## 2. Materials and Methods

### 2.1. Plant Material and Plant Extraction

*E. angustifolium* L. f. *angustifolium* aerial parts were collected by R. Gevrenova and D. Zheleva at the locality “Platoto”, Vitosha Mt., Bulgaria, during the full flowering stage in July 2023. The species taxonomic identity was confirmed by one of us (R. Gevrenova) according to [18]. A voucher specimen was deposited at Herbarium Facultatis Pharmaceuticae Sophiensis, Medical University-Sofia, Bulgaria (Voucher specimen No. 11823). The plant material was dried at room temperature and was extracted with 80% MeOH (1:20 *w*/*v*) by sonication (100 kHz, ultra-sound bath Biobase UC-20C, Jinan, China)) for 15 min (2×) at room temperature, as previously described [21]. The methanol was evaporated in vacuo (40 °C), and water residues were lyophilized (lyophilizer Biobase BK-FD10P, Jinan, China; −65 °C) to yield crude extract (EAE). Then, the lyophilized extract was used for further in vivo and in vitro experiments. The EAE was analyzed by Ultra High-Performance Liquid Chromatography-High Resolution Mass Spectrometry (UHPLC-HRMS, ThermoScientific, Waltham, MA, USA), as previously described [21].

### 2.2. Animals

Twenty-seven male ICR mice with a body weight of 20–30 g were kept in Plexiglas cages (6 per cage) in a 12/12 light/dark cycle under standard laboratory conditions (ambient temperature 20 ± 2 °C and humidity 72 ± 4%). All animals were purchased from the National Breeding Center, Sofia, Bulgaria and allowed a minimum of 7 days of acclimatization before the start of the study. Appropriate food for the strain and fresh drinking water were provided ad libitum. All procedures involving animals were approved by the Animal Care Ethics Committee of the Bulgarian Agency of Food Safety. An ethical clearance (No. 346 of 28 February 2023) was issued.

### 2.3. Experimental Design

The animals were randomly divided into four groups (Table 1), each consisting of six animals. Group 1 included the control animals, which received clean water and food in sufficient quantity throughout the experimental period. Group 2 included the animals from the pathological model with induced colitis. Group 3 were animals with induced colitis that were treated with dexamethasone (DXM, 1 mg/kg). Group 4 were mice with colitis that were orally treated for 3 weeks with *Epilobium angustifolium* extract (EAE) (300 mg/kg). In a preliminary study, a dose of 3000 mg/kg of the extract was administered to 3 mice. No toxic side effects or death were observed two weeks after a single administration; therefore, for the repeated 21-day treatment 1/10 of this dose or 300 mg/kg was chosen. Ulcerative colitis was induced by administering 3% DSS in the drinking water [24] during the first 7 days of the experimental period of all experimental groups except the control group, which had access to clean tap water. During the whole experimental period, the animals from groups 3 and 4 received dexamethasone 1 mg/kg or EAE 300 mg/kg.

The body weight of the mice was measured daily during the first week of colitis induction and twice weekly for the following two weeks. During the experimental period, stool consistency and the presence of blood in the stool were checked daily. The disease activity index (DAI) was assessed using a scoring system described by Chassaing et al. [13]. The DAI was determined by combining the scores for body weight loss, stool consistency, and gross bleeding, divided by 3 (Table 2).

On the 22nd day, after overnight starvation, the animals from all groups were sacrificed after anesthesia with ketamine/xylazine (80 mg/10 kg, i.p.), blood was taken for a hematological analysis, and colons were removed from the appendix to the anus for macroscopic, histological, and biochemical analysis. After the measurement of length and weight, the colons were cut and washed with physiological saline. A portion of the colons was used for biochemical evaluation, and the remaining parallel parts of the colons were fixed in 10% neutral-buffered formalin for histological analyses.

### 2.4. Experimental Methods

#### 2.4.1. Measurement of Hematological Parameters

Whole blood was analyzed by a semi-automated hematological analyzer BC-2800 Vet (Mindray, Shenzhen, China), according to the manufacturer’s instructions. The count of leukocytes (WBC), erythrocytes (RBC, Er), platelets (PLT), amount of hemoglobin (Hb), and hematocrit (Ht) were measured.

#### 2.4.2. Oxidative Stress Markers

Thiobarbituric acid reactive substances (TBARS), expressed as malondialdehyde (MDA) counterparts, were determined as a marker of lipid peroxidation. The method was described by Polizio and Peña [25]. The quantity of MDA was evaluated using a molar extinction coefficient of 1.56 mM^−1^ cm^−1^ and expressed in nmol/g wet tissue. The reduced glutathione (GSH) was quantified by assessing the non-protein sulfhydryls after the precipitation of proteins with trichloracetic acid (TCA), using the method reported by Bump et al. [26]. The absorbance was defined at 415 nm, and the results were indicated as nmol/g wet tissue. The antioxidant enzymes activity was tested in 10% supernatant of colon homogenates prepared in 0.05 M phosphate buffer (pH 7.4). Glutathione peroxidase activity (GPx) was estimated spectrophotometrically at 340 nm using a coupled reaction system consisting of GSH and glutathione reductase and an extinction coefficient of 6.22 mM^−1^ cm^−1^ [27]. CAT activity was assessed as reported by Aebi [28] by assessing H_2_O_2_ degradation through the decrement in absorbance at 240 nm for 1 min. Enzyme activity was calculated using the molar extinction coefficient of 0.043 mM^−1^ cm^−1^ and expressed as nmol/min/mg of protein. The superoxide dismutase activity (SOD) was measured according to the method of Misra and Fridovich [29], following the autoxidation of epinephrine, and using the molar extinction coefficient of 4.02 mM^−1^ cm^−1^. SOD activity was expressed as nmol of epinephrine that was prevented from auto-oxidation after addition of the sample.

The myeloperoxidase activity was assessed spectrophotometrically by the modified method of Sánchez-Fidalgo et al. [30] at 450 nm and expressed as U MPO/mg tissue. The protein content was measured according to the method of Lowry et al. [31].

#### 2.4.3. Histopathological Examination

Histopathological examination of the colons was performed using the method of Bancroft and Gamble [32]. For light microscope evaluation, colon tissues were fixed in 10% buffered formalin, and thin sections (4 μm) were subsequently stained with hematoxylin/eosin for general histological feature determination. The sections were observed under a high-power microscope, and photomicrographs were taken using “Olympus” CX31 (Karl Zeiss, Oberkochen, Germany) and Camera “Olympus x Optical zoom” with objective “PlanaC” 4/0.10 (Karl Zeiss, Oberkochen, Germany).

#### 2.4.4. Statistical Analysis

The statistical program “MEDCALC” version 23.2.1 was used for the analysis of the in vivo data. The results are expressed as mean ± SD for six animals in each group. Comparisons within two groups were made by Student’s *t*-test. One-way analysis of variance (ANOVA) with post hoc multiple group comparisons (Dunnet *t*-test) was used to assess statistical differences. Values of *p* < 0.05 were considered statistically significant.

## 3. Results

UHPLC-HRMS analysis of the secondary metabolites in EAE was previously described [21]. Chromatograms and detailed composition of the extract are presented in Appendix A. Flavonoids were found to be the main classes’ secondary metabolites found in EAE (66.5%), followed by acylquinic acids (16.64%), and gallo- and ellagitannins (10.53%). The UHPLC-HRMS profile was dominated by kaempferol *O*-deoxyhexoside, quercetin 3-*O*-deoxyhexoside, oenothein B, neochlorogenic acid, 3-*p*-coumaroylquinic acid, and ellagic acid [21].

The results of the measured values of the length and weight of the colon in the experimental groups are presented in Table 3. 

The weight of the colon of the model group of animals with induced colitis was 51% less and its length was 35% shorter compared to the colon of the control mice. In the group treated with the extract, the length and weight of the colon remained close to the control values, with the weight being 52% higher and the length being 43% longer compared to the animals with DSS-induced colitis. The values were similar in the dexamethasone-treated group.

DSS-induced colitis is marked by an elevated disease activity index (DAI), which reflects clinical symptoms such as weight loss, changes in stool consistency, and the presence of blood in the feces (hematochezia) (Figure 1). Monitoring body and colon weight loss served as a key indicator of disease progression in this model. Administration of DSS caused a significant decrease in body weight, along with the appearance of loose stools and bloody diarrhea—effects that became especially pronounced after the fourth day of the experiment, compared to the control group (Figure 1). These symptoms, including rectal bleeding and weight loss, were associated with colon shortening, a known consequence of DSS exposure (Figure 2a). The severity of colitis and related complications is inversely related to colon length [33,34]. Among the treated groups, animals given dexamethasone exhibited the least weight loss and minimal intestinal bleeding, particularly during the final week of observation. Visible macroscopic changes in the colon further supported these findings (Figure 2a). Treatment with EAE, and more effectively with dexamethasone, significantly improved all DAI-related parameters (*p* < 0.05) compared to the DSS group (Figure 1 and Figure 2b). Nonetheless, full recovery to baseline (control) values was not achieved by day 21.

Data from the measurement of hematological parameters are presented in Table 4. DSS-induced colitis was characterized by extremely decreased hemoglobin levels, erythrocytes, and hematocrit levels by 66%, 56%, and 32% compared to the controls. At the same time, the level of WBCs significantly increased by 89%, and PLTs modestly decreased by 26% compared with the control group. Three weeks of administration of the EAE led to a significant amelioration of the above-mentioned parameters. Hemoglobin, red blood cells, and hematocrit were increased by 87%, 56%, and 40%, respectively, and WBCs were decreased by 34% compared to the DSS-treated model group. Dexamethasone improved some of the hematological parameters more significantly compared to EAE by increasing HGB by 125% compared to mice treated with DSS alone and by 23% more than the pathological group treated with EAE. At the same time, DXM reduced the leukocyte count to a statistically significant extent by 55% compared to the DSS group and by 31% compared to the EAE treatment group.

Seven-day treatment of animals with DSS resulted in a statistically significant increase in the amount of MDA in colon homogenate by 72% and the activity of MPO by 136% compared to the control (Table 5). In addition, a decreased level of GSH by 47% and decreased activity of CAT, SOD and GPx by 42%, 49%, and 44%, respectively, compared to the control were found. Oral administration of EAE significantly reduced the amount of MDA and the activity of MPO by about 23% compared to the group treated with DSS only. Administration of the extract was also associated with an increased level of GSH by 64% and increased activity of the antioxidant enzymes CAT, SOD, and GPx by 34%, 53%, and 70%, respectively, compared to the pathological group with DSS-induced colitis. Similar results were observed in the group treated with dexamethasone, in which MDA and MPO decreased by 47% and 31%, respectively, compared to the DSS group. GSH, CAT, SOD, and GPx increased by 42%, 52%, 47%, and 48%, respectively, compared to the DSS-treated group. In both groups of animals, the results evaluating biomarkers of oxidative stress and inflammation were almost identical. Only in terms of the amount of MDA did dexamethasone reduce it more significantly compared to EAE (Table 5).

### Histopathological Findings

Figure 3 illustrates the results of the histopathological evaluation. The colon tissue from the control group exhibited a normal structure with intact mucosa and crypt architecture and no evidence of ulceration or mucosal thickening (Figure 3a). In contrast, the pathological model group (Figure 3b) displayed severe colitis, characterized by dense, non-specific inflammatory cell infiltration within the mucosa and beneath the lamina muscularis mucosae. This was accompanied by marked mucosal thickening, substantial epithelial damage, crypt loss, and ulcer formation. Treatment with dexamethasone (Figure 3c) notably reduced inflammation, as indicated by the absence of significant infiltrates in the lamina muscularis mucosae and the presence of elongated intestinal villi with regulated inflammatory activity. Similarly, colonic samples from the group receiving 300 mg/kg of EAE for three weeks (Figure 3d) maintained villi height, with only mild intracellular edema and minimal inflammatory presence in the interstitial tissue.

## 4. Discussion

Drug options for the treatment of ulcerative colitis (UC) are often associated with severe adverse effects, limited bioavailability, and development of resistance that reduces their therapeutic efficacy. There is an urgent need to explore natural strategies as safe and alternative modalities for the management of UC. Currently, approximately 40% of UC patients find relief through natural products, which may help reduce toxic side effects and maintain long-term clinical remission [35].

In recent years, extensive research has focused on the protective effects of various plants and plant-derived compounds against ulcerative colitis (UC). Substances such as polyphenol extracts from green tea, marine-derived bioactive compounds, algae, arbutin, aloe polysaccharides, berberine, curcumin, and Ginsenoside Rh2 have all shown promising results [36,37,38]. Additionally, non-starch polysaccharides (NSPs) sourced from nature, recognized as functional carbohydrates, have demonstrated notable therapeutic potential for UC due to their strong anti-inflammatory and immune-regulating properties [39]. One of the key advantages of natural products lies in their multi-targeted action and diversity of therapeutic effects [38].

The underlying pharmacological mechanisms of these plant-based secondary metabolites have been progressively clarified. Key pathways include enhancement of glutathione metabolism, suppression of the JAK-STAT signaling pathway [40], inhibition of PI3K/AKT signaling [41], and suppression of NLRP3 inflammasome activation [42]. Other effects involve decreased colonic myeloperoxidase (MPO) activity and reduced production of tumor necrosis factor-alpha (TNF-α) [43], as well as downregulation of ICAM-1, nitric oxide (NO) synthesis, and inducible nitric oxide synthase (iNOS) gene expression [44], along with COX-2 inhibition [45].

This study focused on evaluating and comparing the protective effects on the mucosa of *E. angustifolium* extract (EAE) and dexamethasone in mice with colitis induced by DSS. Animal models of inflammatory bowel disease (IBD) are essential for exploring the mechanisms behind IBD development and for testing new treatment approaches. Among these, the DSS-induced colitis model is one of the most commonly used to investigate the causes and possible therapies for ulcerative colitis (UC), a major type of IBD. This model is particularly useful because it is consistently reproducible, easy to create and maintain, and replicates many key aspects of human IBD [34].

When DSS is given, it harms the intestinal lining, causing oxidative stress and inflammation. This leads to the activation of the NF-κB pathway, which promotes inflammation, and the Nrf2 pathway, which supports antioxidant defenses. Examining both pathways in the DSS model provides an important understanding of the interplay between inflammation and cellular protection, closely mirroring the mechanisms seen in human disease [46]. In colitis, oxidative stress remains ongoing for a long time, during which the inflammatory pathways become dominant over the body’s antioxidant defenses. This sustained oxidative stress further intensifies the activation of the NF-κB inflammatory pathway and increases the production of malondialdehyde (MDA), a marker of oxidative damage, resulting in harm to cellular structures [47].

Neutrophil-myeloperoxidase (MPO), a heme-containing enzyme that generates large amounts of hypochlorous acid during inflammation, contributes to the onset and progression of ulcerative colitis (UC). Elevated colonic MPO levels and the presence of neutrophil extracellular traps are closely linked to disease severity in patients with inflammatory bowel disease (IBD). Blocking MPO activity pharmacologically has been shown to reduce disease symptoms in experimental colitis models [48].

Ulcerative colitis involves repeated cycles of inflammation and healing in the lining of the colon and rectum, resulting in tissue damage. This includes changes in the structure of crypts, formation of erosions and ulcers, and the presence of inflammatory cells like neutrophils and eosinophils [49].

All these pathological processes were observed in the present study. DSS significantly increased the amount of MDA in the colon, the activity of MPO, and the number of white blood cells. At the same time, this chemical substance reduced the level of the endogenous antioxidant and cell protector GSH and the activity of antioxidant enzymes compared with the control group. Histopathological changes were also observed. The colitis induced by DSS was accompanied by severe anemia due to bleeding from the colon and a subsequent huge decrease in the level of hemoglobin, erythrocytes, and hematocrit.

Oral administration of the EAE significantly mitigated these disturbances similarly to the positive control dexamethasone, a commonly used glucocorticoid with powerful anti-inflammatory action. The EAE was found to contain a rich variety of beneficial secondary metabolites, such as acylquinic acids, gallo- and ellagitannins, flavonoids, phenolic acids, and their glycosides. A total of 46 compounds was identified for the first time in this species. The total phenolic and flavonoid contents were measured at 85.04 ± 0.18 mg GAE/g and 27.71 ± 0.74 mg QE/g, respectively [21]. The investigated EAE possesses a powerful antioxidant capacity and actively scavenged DPPH and ABTS radicals along with a high reducing and metal chelating power in CUPRAC and FRAP assays [21]. These findings were confirmed in the present in vivo investigation. EAE significantly increased the level of GSH and the activity of antioxidant enzymes, as well as it decreased the quantity of MDA and the activity of MPO compared to the colitis model group. Most probably, these effects are due to the abundant presence of flavonoids and phenolic acids in the herbal extract. Flavonoids possess a number of medicinal benefits, including anticancer, antioxidant, anti-inflammatory, and antiviral properties. Most of the flavonoids are widely accepted as therapeutic agents [50]. They may prove highly valuable in both acute and chronic intestinal inflammation through diverse mechanisms, including protection against oxidative stress and preservation of epithelial barrier function and immunomodulatory properties in the gut [51]. The pharmacological mechanisms linked to the flavonoids and phenolic compounds are related to anti-inflammation, promotion of mucosal healing, maintenance of intestinal immune homeostasis, and regulation of intestinal flora [52,53].

It has been shown that quercetin reduces colon inflammation and damage and contributes to mucosal healing in colitis induced by DSS [54].

Ellagitannins found in the EAE may help suppress oxidative and inflammatory responses, support the integrity of the gut barrier and microbiota, and serve as a dietary means to manage and prevent ulcerative colitis [55]. Despite their low bioavailability, the accumulation of these compounds in the intestinal cavity may be adequate for treating ulcerative colitis, as the disease is confined to the intestinal lining [56]. This effect may stem from the interaction between ellagitannin hydroxyl groups and intestinal mucosal proteins, leading to the formation of a protective layer or partial absorption of tannic acid into the bloodstream, where it can exert its biological effects within the intestine [55]. Ellagic acid, released after hydroxylation of ellagitannins, attenuates the severity of acute DSS-induced colitis in mice and alleviates the trend of liver, spleen, and weight changes in UC mice. It also improves colon oxidative stress, inflammation, and pathological damage in this pathology [57]. Pandurangan et al. found that gallic acid significantly attenuates the DAI and colon shortening and improves the histological alterations caused by DSS [58]. In addition, the authors found a decrease in the level of MDA and increased activities of enzymic antioxidants CAT, SOD, and GPx.

In the present study, we investigated the antioxidant activity of an *E. angustifolium* extract (EAE) in a murine model of DSS-induced colitis. To benchmark its efficacy and mode of action, we conducted a comparative analysis with dexamethasone (DXM), a potent glucocorticoid widely used in the clinical management of inflammatory diseases. Our findings clearly demonstrate the broad-spectrum alleviating effects of EAE on clinical, biochemical, and histological manifestations of DSS-induced colitis in mice, with efficacy comparable to that of DXM. However, as a classical anti-inflammatory and immunomodulatory agent, dexamethasone exerts more definitive effects on some parameters related to this pathological condition. This drug reduces to a greater extent some markers induced by the administration of DSS, namely WBC and MDA, preserves body weight and the HGB level, and reduces intestinal bleeding more significantly than EAE. Histological evaluation confirmed that the EAE effectively mitigated colon shortening, preserving mucosal structure and reducing inflammatory infiltration. Furthermore, EAE restores oxidative balance by reducing MDA and MPO levels, while increasing endogenous antioxidant defenses (GSH, CAT, SOD, and GPx) to the same level as DXM. These findings provide insights into EAE’s potential as a natural, multi-target therapeutic candidate for ulcerative colitis. Its abundant content of flavonoids, phenolic acids, and ellagitannins is likely responsible for its mucosal protective effects, presenting a compelling alternative to traditional corticosteroid therapy.

As highlighted in recent research studies, extracts of the three most popular *Epilobium* species (*E. angustifolium*, *E. hirsutum*, and *E. parviflorum*) have inhibited the activity of hyaluronidase and lipoxygenase [59]. The inhibition of hyaluronidase is related with the presence of oenothein B, a strong inhibitor of this enzyme with IC_50_ of 1.1 M. In addition, the tested extracts inhibited myeloperoxidase release from stimulated neutrophils, and it was found that oenothein B inhibited the enzyme release similarly to indomethacin. The excessively activated neutrophils can contribute to a variety of chronic diseases, such as colon and bowel inflammation. Moreover, the studied *Epilobium* extracts significantly reduced the production of ROS from f-MLP and PMA-induced neutrophils with IC_50_ 5 g/mL and 25 g/mL, respectively. The inhibitory activity of aqueous *Epilobium* extracts on the COX-1 activity may be related to the presence of oenothein B, although rather weak in comparison with the activity towards hyaluronidase and lipoxygenase [59]. Moreover, oenothein B dose-dependently induced a number of phagocyte functions in vitro, including intracellular Ca^2+^ flux, production of ROS, chemotaxis, (NF)-κB activation, and proinflammatory cytokine production [60]. The dimeric hydrolysable tannin showed in vivo activity by stimulating the production of keratinocyte chemoattractant (KC) and promoting neutrophil migration to the peritoneum following intraperitoneal injection. Its capacity to influence phagocyte functions both in vitro and in vivo indicates that oenothein B likely contributes to the therapeutic effects of *E. angustifolium* extracts [60].

Kaempferol and its glycosides are among the dominant constituents of *E. angustifolium* extract. Kaempferol alleviates the gross symptoms of DSS-induced colitis in mice and decreases colonic injury [61]. It also suppresses colon shortening, the DAI, and MPO activity [62]. There are studies that show that kaempferol 3-(2″,4″-diacetylrhamnoside) has a high affinity for binding and inhibiting MPO [63]. Kaempferol and α-rhamnoisorobin were found to reduce NF-κB-driven luciferase activity. Both compounds also decreased iNOS mRNA expression in a dose-dependent way. These findings indicate that their ability to inhibit nitric oxide (NO) production is likely due to the downregulation of iNOS mRNA [64]. Treatment with kaempferol led to a significant decrease in blood levels of various inflammatory markers, including TNF-α, leukocytes, cytokines, IL-1β, intracellular adhesion molecule-1, and E-selectin [65]. The flavonoid effectively disrupted STAT3 transactivation and further suppressed the activation of inflammatory cytokines [66]. Kaempferol exhibited strong inhibitory effects on COX-1 and COX-2 enzymes in cell-free in vitro assays [67]. It also reduced COX-2 expression by blocking Src-kinase activity triggered by UVB exposure, thereby potentially impacting other disease-related processes [68]. Previous studies have shown that kaempferol can suppress the proliferation of both unstimulated and IL-1β-stimulated rheumatoid arthritis synovial fibroblasts (RASFs). It also reduces IL-1β-induced production of MMP-1, COX-2, MMP-3, and PGE2. Furthermore, kaempferol has been found to inhibit the activation of NF-κB and the phosphorylation of ERK-1/2, p38, and JNK—key proteins involved in the inflammatory response in rheumatoid arthritis (RA). These findings suggest that kaempferol holds promise as a potential therapeutic agent for RA treatment [69]. It has been demonstrated that myricetin 3-*O*-glucuronide from *E. angustifolium* leaves exerts significant anti-inflammatory activity at micromolar range and has an impact on the protective effect of the extract along with oenothein [19]. Zhao et al. discovered that myricetin decreased the production of nitric oxide (NO), MPO activity, and MDA quantity, while increasing the activity of SOD and GPx in acute experimental colitis induced by DSS [70].

Despite the limited bioavailability of the acylquinic acids, transformations that occur in the colon facilitate their absorption as microbial catabolites (dihydrocaffeic acid, ferulic acids, and their sulfates and glucuronides) [71]. It has been shown that dihydrocaffeic acid significantly reduced secretion of the proinflammatory cytokines TNF-α, IL-1b, and IL-6. Indeed, some gut flora catabolites of caffeoylquinic acids and their phase II metabolites may have anti-inflammatory activity [72].

Overall, most of the biactive compounds identified in EAE, when administered alone in mouse models of induced colitis, exhibited protective effects and reduced the damage caused by DSS. This identifies EAE as a promising natural agent for colon protection in inflammatory diseases.

## 5. Conclusions

The results of this study highlight that *E. angustifolium extract* (EAE) exhibits effects comparable to dexamethasone in alleviating DSS-induced colitis. Both treatments significantly improved colon weight and length, reduced disease activity, and ameliorated hematological and oxidative stress parameters. EAE treatment similarly reduced MDA levels and MPO activity and increased antioxidant enzyme activities (CAT, SOD, and GPx) and GSH levels, matching the effects produced by dexamethasone. Histopathological analysis also showed that both treatments effectively reduced inflammation and preserved colon structure. Our findings suggest that *E. angustifolium* extract (EAE) holds potential as a natural therapeutic candidate for the management of IBD, demonstrating efficacy comparable to that of conventional pharmacological treatments. However, to fully validate these effects and facilitate its development as a therapeutic product, further in-depth studies are necessary to elucidate the underlying molecular mechanisms of its anti-inflammatory activity and optimize its application in future clinical settings.

## Figures and Tables

**Figure 1 cimb-47-00444-f001:**
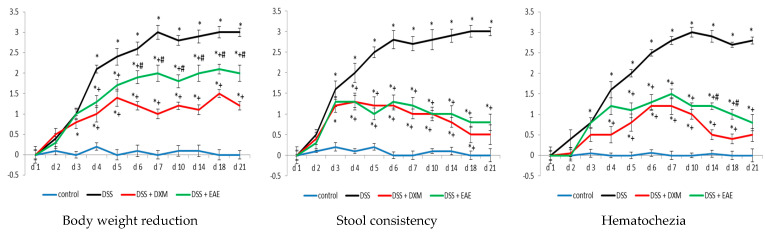
Daily assessment of body weight reduction, stool consistency, and presence of blood in the feces in the experimental groups. Results are expressed as mean ± SD (n = 6). * *p* < 0.05 vs. control; ^+^ *p* < 0.05 vs. DSS; ^#^ *p* < 0.05 vs. DSS + DXM.

**Figure 2 cimb-47-00444-f002:**
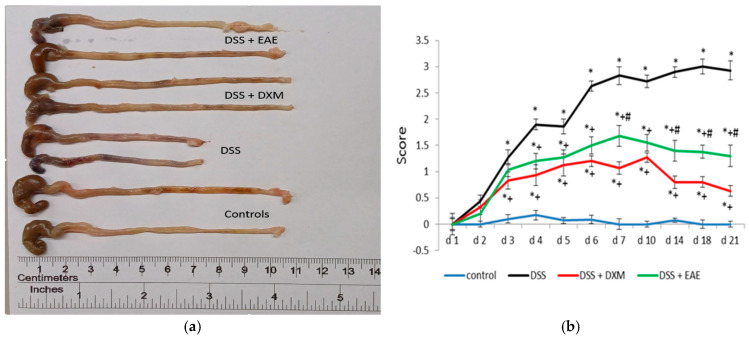
Colon length of the experimental groups (**a**) and disease activity index (DAI) scores (**b**) at the end of the animal trial. Results are expressed as mean ± SD (n = 6). * *p* < 0.05 vs. control; ^+^ *p* < 0.05 vs. DSS; ^#^ *p* < 0.05 vs. DSS + DXM.

**Figure 3 cimb-47-00444-f003:**
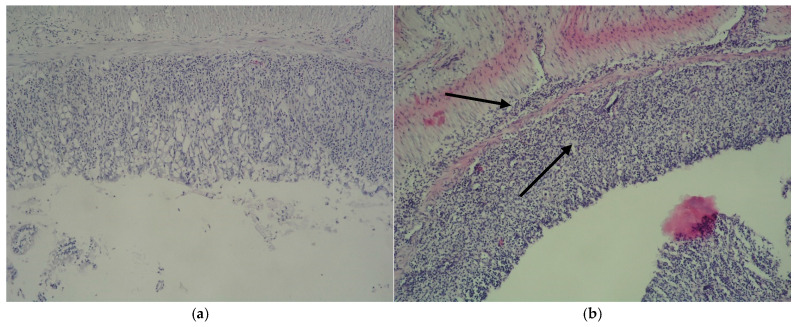
Histopathological micrographs. (**a**) Colon from control mice; (**b**) DSS-induced colitis with inflammatory infiltrations and thickening of the mucosa with severe epithelial damage (arrows); (**c**) colon from mice with colitis treated with dexamethasone, lamina muscularis mucosae without pronounced inflammation (arrow); (**d**) colon from mice with colitis orally treated with EAE 300 mg/kg with scant inflammatory infiltrations.

**Table 1 cimb-47-00444-t001:** Experimental design.

№	Groups (n = 6)	Week 1	Week 2	Week 3
1.	Control	Food and water ad libitum	Food and water ad libitum	Food and water ad libitum
2.	DSS	DSS 3%	Pure tap water	Pure tap water
3.	DSS + DXM	DSS 3% + DXM 1 mg/kg	DXM 1 mg/kg	DXM 1 mg/kg
4.	DSS + EAE	DSS 3% + EAE 300 mg/kg	EAE 300 mg/kg	EAE 300 mg/kg

**Table 2 cimb-47-00444-t002:** Assessment of disease activity index (DAI) score.

Score	Weight Loss (% of Initial wt)	Bleeding	Stool Consistency	Inflammatory Score (DAI)
0	none	Negative	Regular	Normal
1	1–5%	Slight bleeding	Soft unformed excrement	Slight inflammation
2	6–10%	Moderate bleeding	Loose feces	Moderate inflammation
3	>11%	Severe bleeding	Watery diarrhea	Severe inflammation

**Table 3 cimb-47-00444-t003:** Colon weight and length of experimental groups.

Parameters	Controls	DSS 3%	DSS + DXM	DSS + EAE
Colon weight (g)	0.86 ± 0.032	0.42 ± 0.048 *	0.72 ± 0.039 *^+^	0.64 ± 0.022 *^+^
Colon length (cm)	11.1 ± 0.34	7.2 ± 0.52 *	10.8 ± 0.14 ^+^	10.3 ± 0.4 ^+^

Results are expressed as mean ± SD (n = 6). * *p* < 0.05 vs. control; ^+^ *p* < 0.05 vs. DSS. Abbreviations: DSS, Dextran sulfate sodium; DXM, dexamethasone; EAE, *E. angustifolium* extract. Treatment: as described in the Section 2.3.

**Table 4 cimb-47-00444-t004:** Hematological parameters in the experimental groups.

Hematological Parameters	Controls	DSS 3%	DSS + DXM	DSS + EAE
WBC (×10^9^)/L	7.51 ± 1.32	14.22 ± 1.10 *	6.41 ± 1.4 ^+^	9.33 ± 0.8 *^+#^
RBC (×10^12^)/L	9.6 ± 0.70	5.2 ± 0.10 *	8.9 ± 0.9 ^+^	8.1 ± 1.2 ^+^
HGB g/L	140.7 ± 2.4	47.2 ± 5.20 *	106.6 ± 3.8 *^+^	86.6 ± 4.3 *^+#^
HCT %	44.4 ± 2.20	30.2 ± 1.10 *	40.6 ± 3.22 ^+^	42.3 ± 2.06 ^+^
PLT (×10^9^)/L	734.8 ± 64.4	547.3 ± 55.1 *	622.2 ± 36.8 ^+^	812.2 ± 36.2 *^+#^

Results are expressed as mean ± SD (n = 6). * *p* < 0.05 vs. control; ^+^ *p* < 0.05 vs. DSS; ^#^ *p* < 0.05 vs. DSS + DXM. Treatment: as described in the Section 2.3.

**Table 5 cimb-47-00444-t005:** Biomarkers of oxidative stress and inflammation.

Parameters	Controls	DSS 3%	DSS + DXM	DSS + EAE
MDA nmol/g tissue	4.78 ± 0.45	8.22 ± 0.48 *	4.32 ± 0.43 ^+^	6.18 ± 0.62 *^+#^
GSH nmol/g tissue	7.22 ± 0.51	3.81 ± 0.24 *	5.42 ± 0.41 *^+^	6.24 ± 0.32 *^+^
MPO U/mg tissue	0.38 ± 0.04	0.91 ± 0.09 *	0.62 ± 0.07 *^+^	0.72 ± 0.08 *^+^
CAT nmol/mg/min	5.62 ± 0.41	3.24 ± 0.32 *	4.93 ± 0.46 ^+^	4.33 ± 0.28 *^+^
SOD nmol/mg/min	0.62 ± 0.03	0.38 ± 0.02 *	0.56 ± 0.03 ^+^	0.58 ± 0.04 ^+^
GPx nmol/mg/min	1.65 ± 0.12	0.92 ± 0.06 *	1.36 ± 0.09 ^+^	1.57 ± 0.07 ^+^

Results are expressed as mean ± SD (n = 6). * *p* < 0.05 vs. control; ^+^ *p* < 0.05 vs. DSS; ^#^ *p* < 0.05 vs. DSS + DXM. Treatment: as described in the Section 2.3.

## Data Availability

The original contributions presented in the study are included in the article.

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
