# Peer review of "Ulceroprotective Effects of *Epilobium angustifolium* Extract in DSS-Induced Colitis in Mice"

_cimb, 2025, doi:10.3390/cimb47060444_

Round 1
Reviewer 1 Report
Comments and Suggestions for Authors
I am commenting only on the issue of botanical sourcing, as I have no expertise relevant to the study model. It is commendable that the test material was obtained from whole plants, a voucher was prepared, and the locality, plant parts, and stage of development were all specified.
However, I think a few more words about the plant's identification would have been useful. There are about a dozen species (and a common hybrid) of Epilobium in Bulgaria. At least one or two species look similar enough to E. angustifolia that people might be able to confuse them. Since none of the authors are identifiable from their affiliations as botanists, you might report both who collected the plant and who identified it. The number given to the voucher by the herbarium is reported, but not the collector's name and number, which should also be included if possible. If you used any literature to help with that identification, you might cite it.
Author Response
Reviewer 1
I am commenting only on the issue of botanical sourcing, as I have no expertise relevant to the study model. It is commendable that the test material was obtained from whole plants, a voucher was prepared, and the locality, plant parts, and stage of development were all specified.
However, I think a few more words about the plant's identification would have been useful. There are about a dozen species (and a common hybrid) of Epilobium in Bulgaria. At least one or two species look similar enough to E. angustifolia that people might be able to confuse them. Since none of the authors are identifiable from their affiliations as botanists, you might report both who collected the plant and who identified it. The number given to the voucher by the herbarium is reported, but not the collector's name and number, which should also be included if possible. If you used any literature to help with that identification, you might cite it.
Response: Thank you for the valuable comment. Plant description was added in the Introduction section. Additional information about the collection and identification was added in Materials and Methods section.
The plant identification was performed according to Kuzmanov, B., 1979. Epilobium angustifolium L. In: Flora Republicae Bulgaricae, Vol. VII, Onagraceae in: Bulgarian Academy of Sciences (Eds.), pp. 449-454.
E. angustifolium is a perennial herb with creeping rhizome. Stem is 50 to 150 cm high, simple, rarely branched, often reddish. Leaves are thin, usually alternate, oblong-lanceolate, 4-15 cm long, 0.4-3.5 cm wide, with convex, reticulately branched pale lateral veins below, sessile or on short petioles. Inflorescence is a long raceme. Flowers are slightly irregular; petals 10-15 mm long and 5-8 mm wide, violet-pink. Capsule is 4-8 cm long, often reddish. Seeds are pale brown, oblong, narrowed at both ends. Lines 78-84.
Reviewer 2 Report
Comments and Suggestions for Authors
The study investigated the potential ulcer protective effect of Epilobium angustifolium extract in a mouse model. There are some modifications that the authors are advised to take into consideration when reviewing the manuscript:
- Line 23 "Histopathological analysis confirmed EAE’s efficacy in attenuating colonic injury and inflammation, along with improving blood parameters hemoglobin, erythrocytes, and hematocrit". Please re-edit this sentence as it gives the impression that it was the histopathological analysis that showed improved blood parameters, hemoglobin, red blood cells, and hematocrit.
- It is preferable to add a final conclusion to the study at the end of the abstract section.
- In lines 76-90, several bioactive components of the plant are mentioned. Please relate these components to the ulcer-protective effect to illustrate the rationale behind the study of the plant.
- It is best to clearly add the purpose of the study to the end of the introduction section.
- The number of mice used was 6 in each group and 4 groups were studied, a total of 24 mice. Why were 27 mice used (line 107)?
- In the Materials and Methods section, please add the reference used to determine the dose and duration of DSS.
- In Table 3, remove the word "in" from “colon weight in” and “colon length in”.
- How was the weight loss of individual mice assessed as each group was placed in a cage? Were the mice, for example, marked? What method was used for marking?
- In the results section, the comparison between the effects of the plant extract and dexamethasone is unclear on most measures. To what extent did the plant extract approach the drug from its effect?
- In the discussion section, further linkage between the effect of the plant extract on all studied parameters and its identified components is needed.
- Please ensure that no information is provided without a scientific reference. For example, the sentence "It has been shown that dihydrocaffeic acid significantly reduced the secretion of the proinflammatory cytokines TNF-α, IL-1b, and IL-6. Indeed, some gut flora catabolites of caffeoylquinic acids and their phase II metabolites may have anti-inflammatory activity", please add the reference.

Author Response
Reviewer 2
The study investigated the potential ulcer protective effect of Epilobium angustifolium extract in a mouse model. There are some modifications that the authors are advised to take into consideration when reviewing the manuscript:
- Line 23 "Histopathological analysis confirmed EAE’s efficacy in attenuating colonic injury and inflammation, along with improving blood parameters hemoglobin, erythrocytes, and hematocrit". Please re-edit this sentence as it gives the impression that it was the histopathological analysis that showed improved blood parameters, hemoglobin, red blood cells, and hematocrit.
Response: Thank you for pointing out the inaccuracy. It was re-edited as follows: Histopathological analysis confirmed EAE’s efficacy in attenuating colonic injury and inflammation. Blood parameters hemoglobin, erythrocytes, and hematocrit were also improved. Lines 23-25.
2. It is preferable to add a final conclusion to the study at the end of the abstract section.
Response: Thank you for the valuable suggestion. Conclusion was done as follows: “Our study shows that EAE has potential as a natural therapeutic candidate for the treatment of UC, demonstrating efficacy comparable to that of conventional pharmacological treatments” lines 25-27
3. In lines 76-90, several bioactive components of the plant are mentioned. Please relate these components to the ulcer-protective effect to illustrate the rationale behind the study of the plant.
Response: Additional information about the correlation between bioactive compounds in E. angustifolium extract and ulcer-protective effect was provided (See Introduction and Discussion sections).
4. It is best to clearly add the purpose of the study to the end of the introduction section.
Response: Thank you for the valuable suggestion. It was added as follows: “The purpose of the present study was to evaluate the ulcer protective effects of EAE in mice with DSS-induced colitis and to compare these effects with the positive control dexamethasone, a widely used corticosteroid in the clinical practice.” Lines 109-111
5. The number of mice used was 6 in each group and 4 groups were studied, a total of 24 mice. Why were 27 mice used (line 107)?
Response: In the Experimental design section was stated that:” In a preliminary study, a dose of 3000 mg/kg of the extract was administered to 3 mice.” So, the "missing" 3 mice were used to determine the toxicity of the extract.
6. In the Materials and Methods section, please add the reference used to determine the dose and duration of DSS.
Response: Thank you for the valuable suggestion. The reference № 24 was added.
7. In Table 3, remove the word "in" from “colon weight in” and “colon length in”.
Response: Thank you. It was done.
8. How was the weight loss of individual mice assessed as each group was placed in a cage? Were the mice, for example, marked? What method was used for marking?
Response: Thank you for the questions. The four groups of animals were marked with four different colors of non-toxic laboratory animal markers. In our laboratory we have an official animal marking scheme from 1 to 10. Marking starts from the front right paw, clockwise, as shown in the diagram below, No. 10 remains a white, unmarked animal. In the places shown, the animal is simply lightly marked, the number is not written, since it is known which number the corresponding marking corresponds to. In this way, we also reduce the size of the marking, although the markers are not toxic to the animals.
9. In the results section, the comparison between the effects of the plant extract and dexamethasone is unclear on most measures. To what extent did the plant extract approach the drug from its effect?
Response: Thank you for the valuable question. An additional statistical analysis was performed comparing the groups treated with dexamethasone and those treated with EAE. Statistically significant differences in the results are marked with the # sign. Overall, the results evaluating the biomarkers of oxidative stress and inflammation in both groups of animals were almost identical. Only in terms of the amount of MDA, dexamethasone reduced it to a greater extent than EAE. Dexamethasone also had more pronounced effects in terms of body weight reduction, disease activity index, more significantly reduced the number of leukocytes and increased the level of hemoglobin.
These comparisons are noted in the results section, as well as in the discussion.
- In the discussion section, further linkage between the effect of the plant extract on all studied parameters and its identified components is needed.
Response: Additional information was provided (See Discussion section). Lines: 389-390; 398-406; 448-451; 471-474.
11. Please ensure that no information is provided without a scientific reference. For example, the sentence "It has been shown that dihydrocaffeic acid significantly reduced the secretion of the proinflammatory cytokines TNF-α, IL-1b, and IL-6. Indeed, some gut flora catabolites of caffeoylquinic acids and their phase II metabolites may have anti-inflammatory activity", please add the reference.
Response: Thank you for the valuable comment. The reference № 72 was added. Line 480.

Round 2
Reviewer 2 Report
Comments and Suggestions for Authors
The proposed amendments have been made and the questions raised have been answered.